# FAST4D—A New Score to Reduce Missed Strokes in Emergency Medical Service: A Prospective, Multicentric Observational Proof-of-Concept Trial

**DOI:** 10.3390/jcm13175033

**Published:** 2024-08-25

**Authors:** Christian Claudi, André Worm, Donata Schmohl, Martin Juenemann, Omar Alhaj Omar, Hendrik Loesche, Hagen B. Huttner, Patrick Schramm

**Affiliations:** 1Department of Neurology, University Hospital of the Justus-Liebig-University Giessen, 35385 Giessen, Germany; 2Department of Emergency Medicine, Campus Lippe, University Hospital of Bielefeld, 33615 Bielefeld, Germany; 3Emergency Medical Service Educational Center of Malteser, District of Hesse, Rhineland Palatinate and Saarland, 35578 Wetzlar, Germany

**Keywords:** stroke, emergencies, emergency medical service, triage, diagnostic techniques, neurological, paramedics, emergency medical technicians

## Abstract

**Background**: Undoubtedly, overlooking a stroke can result in severe disability or even death. However, identifying stroke patients in the prehospital setting poses a significant challenge. While the Face–Arm–Speech–Time (FAST) score is widely used, its effectiveness has been questioned because of its focus on symptoms primarily associated with anterior circulation strokes. In response to this limitation, we developed the innovative FAST4D score and conducted a comparative analysis of stroke detection rates between the novel FAST4D score and the FAST score. **Methods**: This prospective, multicenter proof-of-concept study aimed to assess stroke detection rates using both the FAST score and the new FAST4D score, which incorporates additional items such as the acute onset of diplopic images, deficit in the field of vision, dizziness/vertigo, and dysmetria/ataxia. Following their presentation to emergency medical services, all patients suspected of having a stroke and those diagnosed with a stroke upon discharge were included in this study. The diagnostic performance of the novel FAST4D score was evaluated and compared with that of the FAST score. **Results**: Between May 2019 and June 2021, a total of 1469 patients (749 female) were enrolled, with 1035 patients discharged with the diagnosis of stroke. Notably, 259 patients were identified solely through the FAST4D score. This resulted in a significantly higher rate of correctly identified as having had a stroke (stroke detection rate, sensitivity) with the new FAST4D score (93%) compared with the established FAST score (78%) (*p* < 0.001). This resulted in a reduction in false negative diagnoses by 65%. **Conclusions**: The novel FAST4D score demonstrated a 15-percentage increase in the stroke detection rate. This heightened detection rate holds the potential for more accurate patient allocation to stroke units, consequently reducing the time to revascularization.

## 1. Introduction

### 1.1. Background

The adage “time is brain” [1] has long served as a cornerstone principle in stroke care worldwide. However, despite widespread awareness, the comprehensive recognition of all common stroke symptoms in the emergency setting remains a formidable challenge. This principle underscores the urgent need for prompt diagnosis and therapy to mitigate the progressive and irreversible neuronal damage following a stroke event. Nonetheless, the variable nature of stroke symptoms complicates their immediate recognition, underscoring the complexity of stroke assessment [2].

In the majority of cases, stroke patients first encounter emergency medical service (EMS) professionals [2,3]. Hence, the early identification of stroke symptoms is paramount for EMS personnel [4]. Various scoring systems have been devised to aid in the recognition of key stroke symptoms [5,6]. Among these, the FAST (Face–Arm–Speech–Time) score has gained widespread acceptance among EMS personnel and laypersons alike. However, upon critical examination, it becomes evident that the FAST score primarily focuses on symptoms associated with anterior circulation strokes, potentially overlooking those related to posterior circulation strokes [7]. Jones et al. observed that a significant proportion of patients, approximately 25%, present with symptoms not encompassed by the FAST score [2]. This deficiency is particularly notable in posterior circulation strokes, where symptoms are often less specific and may not readily align with typical stroke presentations [8,9,10,11,12].

To address this gap, alternative scoring systems such as “BE-FAST” have been proposed, incorporating additional symptoms such as balance disturbances and visual impairment. However, subsequent studies have yielded inconclusive results regarding the efficacy of BE-FAST in enhancing stroke detection [13,14].

In light of these findings, we aimed to enhance stroke detection by augmenting the FAST score with specific symptoms characteristic of posterior circulation strokes. These include eye motility disorders, visual field defects, dizziness/vertigo, and ataxia. Eye motility disorders, attributed to brainstem involvement, may elude detection by inexperienced examiners [15], potentially leading to the under-prioritization of patients. However, diplopic images resulting from eye muscle palsy offer a more discernible manifestation. Visual field defects may serve as critical indicators of posterior circulation, as mentioned in the BE-FAST trial above. Likewise, sudden onset dizziness or vertigo, while nonspecific, can be crucial and occasionally the sole indicator of cerebellar infarctions in up to 10% of cases [16].

Studies have demonstrated that training EMS personnel in the finger–nose test significantly enhances the detection rate of posterior circulation strokes from 46% to 74%, further underscoring the importance of recognizing ataxia as a hallmark symptom [17].

To streamline the inclusion of these additional symptoms, all commencing with the letter “D” (diplopic images, deficit in the field of vision, dizziness/vertigo, and dysmetria/ataxia), we expanded the FAST acronym, thus giving rise to the novel FAST4D score.

This simple addition of four new items, tailored specifically for posterior circulation strokes, has the potential to improve the identification of these strokes and warrants further investigation, as outlined in the present study.

### 1.2. Aim

In this study, our main objective is to assess and compare the effectiveness of stroke detection between the novel FAST4D score and the conventional FAST score. Additionally, we aim to evaluate the posterior stroke detection rate. Furthermore, we investigate the distribution and frequency of the newly introduced 4D items.

## 2. Materials and Methods

### 2.1. Study Population

This study is a prospective, multicenter observational cohort study conducted in 2 districts in Germany, with a combined population of approximately 565,000 inhabitants. Patient recruitment primarily took place at the general hospitals in Wetzlar and Friedberg, both of which have a department of neurology, including a certified stroke unit, and a department of general medicine. The stroke units accommodate a comparable number of stroke patients for treatment. As thrombectomy is not performed in these hospitals, patients with large vessel occlusions (LVOs) are usually transferred to the nearby Giessen University Hospital for endovascular procedures. Data collection and pseudonymization were prospectively conducted at both hospitals, adhering to the principles outlined in the Declaration of Helsinki. Ethics approval for this study was obtained from the Ethics Committee of the Department of Medicine at Justus Liebig University in Giessen, Germany, on 18 January 2019 (Approval number: 215/18).

### 2.2. Study Design and Data Collection

All patients who were either brought to the emergency department (ED) by emergency medical services (EMSs) or who presented themselves at the ED with a suspected stroke were included in the data collection. A patient was defined as having a suspected stroke if any of the FAST4D items were positive. This criterion was explicitly stated in the standard operating procedures (SOPs) of both the EMSs and the ED. The FAST4D score was documented either by paramedics at the emergency scene as part of the EMS protocol or by the nursing staff in the ED upon admission during the triage process.

The inclusion criteria included the following (concise list):Patient Presentation.
Patients who were brought to the emergency department (ED) by emergency medical services (EMSs).Patients who presented themselves at the ED with suspected stroke.Patients who were either brought to the ED by EMSs or self-admitted without a suspected stroke but were later diagnosed with a stroke during clinical evaluationSuspected Stroke Definition.
A patient was considered to have had a suspected stroke if any of the FAST4D items were positive.Documentation.
FAST4D score was documented by paramedics at the emergency scene (as part of the EMS protocol).Alternatively, the FAST4D score was documented by nursing staff in the ED during admission triage.

As the items of the FAST4D score were individually recorded and already encompassed the items of FAST, a direct comparison between FAST and FAST4D scores was possible using the same dataset. To calculate test quality criteria such as specificity and sensitivity, it was necessary to include all potential stroke patients, even those who were not initially referred with suspected stroke but were ultimately diagnosed with stroke upon discharge (false negative patients). Data for these cases were obtained from patients’ medical records. The reporting of results follows the Strengthening the Reporting of Observational Studies in Epidemiology (STROBE) Statement [18].

### 2.3. The Novel FAST4D Score

The FAST score, comprising the items Face, Arm, and Speech, has already been utilized and documented in both the EMS and ED settings. In 2019, the FAST4D score, incorporating additional items such as diplopic images, deficit in the field of vision, dizziness/vertigo, and dysmetria/ataxia, was introduced into the EMS protocol within both districts. Paramedics were instructed to replace the use of the FAST score alone with the FAST4D score, as outlined in the updated standard operating procedure for stroke treatment and underlined with a pocket card for each paramedic and the nursing staff (refer to the EMS pocket card depicted in Figure 1). This implementation occurred prior to the commencement of data collection.

The parameters “FAST-positive” and “FAST4D-positive”, along with their respective positive items from the “4D” criteria, were directly extracted from the documentation of the EMSs and ED. The “FAST-positive” parameter was marked as true if at least one item from the FAST score was documented as positive. Similarly, “FAST4D” was flagged as true if any one of the FAST4D score items was documented as positive.

The binary parameter “stroke” served as the primary metric and, along with the subsequent subparameters, was defined based on the discharge diagnosis. Both stroke entities—cerebral ischemia and intracerebral hemorrhage (ICH)—were aggregated and assessed as true. Patients with subarachnoid hemorrhage (SAH) and transient ischemic attack (TIA) were excluded. Stroke and ICH diagnoses were confirmed using CT or MRI scans.

### 2.4. Outcomes

All patients with at least one positive item identified in the scores and subsequently discharged with a diagnosis of stroke were deemed to have been correctly identified as stroke cases by either the FAST score or the FAST4D score.

### 2.5. Statistical Analysis

This study was planned as a prospective observational study over a period of 24 months, with an average number of treated stroke patients per center of 750, theoretically reaching a sample size of 2500 patients. Calculating the sample size with a low effect size of 0.1, an alpha of 0.05, and a power of 0.9, yields a minimum required number of patients of 1051 (G*Power χ^2^ test goodness-of-fit, version 3.1.9.4, University of Duesseldorf, Germany). The minimum number of patients should have thus been reached within the planned timeframe. Descriptive statistics were employed to characterize the study population, and comparisons were conducted using a *t*-test. The test quality of the FAST score and FAST4D score was assessed using 4-field panels with χ^2^ calculation. The detection rate between both tests was analyzed using the McNemar test. Differences in sensitivity and specificity between the FAST score and FAST4D score were computed. Patients with missing data were excluded from the final analysis because of the proof-of-concept approach.

Data analysis was performed using the Statistical Package for the Social Sciences (SPSS) from IBM (version 25, IBM Corporation, New York, NY, USA).

## 3. Results

### 3.1. Patient Inclusion and Characteristics

Data were collected between May 2019 and June 2021. Initially, 1940 patients were screened. However, 471 patients were excluded from the final analysis because of transient ischemic attack, subarachnoid hemorrhage, traumatic brain injury, stroke diagnosis without MRI or CT, or incomplete datasets. Consequently, a total of 1469 patients were included in the final analysis.

Of the patients included in this study, 749 (51%) were female. The mean age was 72 ± 15 years. Among the entire population, 1035 patients suffered a stroke. The distribution of stroke entities was as follows: 963 (93%) had an ischemic stroke and 72 (7%) had an intracerebral hemorrhage (ICH). The diagnosis was made in 274 patients (26.5%) via computed tomography (CT), and in 761 patients (73.5%) through magnetic resonance imaging (MRI). The location of the stroke was in the anterior circulation for 720 (70%) patients, while 261 (30%) patients suffered a posterior circulation stroke. For detailed patient characteristics, please refer to Appendix A in the Appendix A.

### 3.2. Stroke Detection by the Two Scores

In the direct comparison between the traditional FAST score and the new FAST4D score, the stroke detection rate increased from 78% when using the FAST score alone to 93% when employing the FAST4D score (*p* < 0.001) among all 1035 patients diagnosed with stroke. This resulted in the identification of an additional 259 (15.4%) stroke patients.

Out of the 1469 included patients, at least one item of the new FAST4D score was positive in 1400 cases, and among them, 966 were ultimately diagnosed with stroke (including ICH). The sensitivity of the new FAST4D score in detecting stroke was 93.3% (χ^2^ = 30.4; *p* < 0.001; see Table 1).

The traditional FAST score yielded positive results in 1017 out of the 1469 patients included in this study. Among them, 807 patients were diagnosed with a stroke (including ICH). The sensitivity of the FAST score in detecting stroke was 78.0% (χ^2^ = 125.6; *p* < 0.001; see Table 2).

### 3.3. Detection of Posterior Circulation Strokes

In the direct comparison, the posterior stroke detection rate increased from 175 (56%) using the FAST score alone to 298 (95%) using the FAST4D score (*p* < 0.001), encompassing all 314 posterior circulation strokes. This signifies an additional 123 detected stroke patients with the FAST4D score when compared with the FAST score, with only 16 posterior circulation stroke patients missed. Further analysis of FAST and FAST4D regarding only posterior circulation strokes is presented in the Appendix A.

### 3.4. Frequency and Distribution of 4D Items

In the context of this study, the four new items of the FAST4D score were evaluated individually and could therefore be examined separately in terms of their occurrence in strokes. The frequency and distribution of the individual 4D items are presented in Table 3.

## 4. Discussion

The new FAST4D score notably enhances stroke detection by about 15% compared with the traditional FAST score. This improvement leads to a 65% reduction in false negative diagnoses compared with the established FAST score, potentially optimizing the timely allocation of stroke patients to time-dependent treatment of revascularization.

The under-detection of posterior circulation strokes by conventional scoring systems is a well-documented issue, see refs. [2,9,11,16], prompting the development of scoring tools integrating additional parameters, such as FAST4D.

In a retrospective data analysis conducted at the Stroke Center of the University of Kentucky (USA) in 2014, 736 stroke patients were retrospectively examined for symptoms presented upon admission. Among these patients, 14% exhibited no symptoms identified by the FAST score. However, by incorporating “balance” (indicating gait uncertainty or leg weakness) and “eyes” (indicating vision loss or double vision) as additional parameters, only 4% of patients would have gone undetected as stroke cases. These additional items were already integrated into training programs under the mnemonic BE-FAST (Balance–Eyes–Face–Arms–Speech–Time), as reported by the authors [13]. Subsequently, a prospective study investigating BE-FAST involving 359 patients, 159 of whom had experienced a stroke, was conducted. The BE-FAST items were more frequently positive in stroke patients compared with the 200 patients without stroke. However, only facial paralysis and arm weakness emerged as independent variables for predicting stroke detection. In comparison to FAST, the authors concluded that BE-FAST did not enhance stroke detection [14]. This implies that the inclusion of only two additional items may not be adequate for accurately identifying patients with posterior circulation strokes.

In a 2019 study conducted in Michigan, USA, paramedics underwent training in the finger–nose test to detect dysmetria/ataxia. That study compared the detection rates of posterior circulation strokes before and after the training, alongside those of a control group. The results demonstrated a notable rise in the detection rate of posterior circulation strokes from 46% to 74% following the training, while no similar enhancement was observed in the control group [17]. The finger-to-nose test has indeed increased the detection rate of strokes, but it has not become the standard in emergency medical services. Furthermore, it seems insufficient as a standalone additional item. In FAST4D, this item has therefore been integrated along with other items.

Another study augmented the FAST protocol with additional criteria including visual disturbance, ataxia, and vertigo. That investigation revealed a noteworthy enhancement in the sensitivity for detecting posterior circulation strokes [7]. Since these symptoms in combination were not previously included in any score, they formed the basis for the new FAST4D score.

BE-FAST, the mnemonic increasingly utilized in emergency medical services to address posterior circulation strokes, did not result in a notable enhancement in stroke detection. Moreover, the sample sizes in previous studies investigating BE-FAST were notably smaller compared with the current investigation, particularly in the prospective observation comparing BE-FAST with FAST [13,14]. FAST4D was able to demonstrate that the combination of four items clinically improves stroke detection rates in a large prospective study. The enhanced detection rate of strokes in the posterior circulation contributed to an overall improvement in the detection rate of all stroke patients, potentially leading to more timely treatment for these individuals.

Approximately 30% of the strokes investigated in this study were classified as posterior circulation strokes, which are expected to be better identified by the FAST4D score. Compared with the FAST score, the sensitivity of detecting posterior circulation strokes increased by approximately 40 percentage points, rising from 56% to 93% with the implementation of the new FAST4D score.

Furthermore, while the increased detection rate associated with the FAST4D score results in more patients being incorrectly identified as having had a stroke, known as over-triage, the benefits of accurately identifying stroke patients outweigh the drawbacks. Over-triage, though requiring careful assessment, is deemed necessary in emergency medicine to ensure prompt access to recanalization therapy in order to minimize cerebral damage. In the present study, 434 patients with a positive FAST4D score did not have a stroke, of which 210 were also positive on the FAST score. Therefore, an additional 224 patients were incorrectly identified as having had a stroke by incorporating the 4D items into the FAST score (see Appendix A in the Appendix A). These over-triaged patients had primarily neurological diseases and were therefore assigned to the appropriate clinical department where they could then be quickly re-triaged and thus did not lead to an excessive burden on resources. Please find more information in the Appendix A. From the authors’ point of view, such a small over-triage compared with recanalizing therapy for a stroke patient, which is no longer possible because of misallocation, is absolutely necessary from ethical, clinical, and ultimately economic points of view.

Regarding the specific items of the FAST4D score, while diplopic images, deficits in the field of view, and dysmetria are significantly more prevalent in stroke patients, dizziness/ vertigo is notably more frequent in non-stroke patients. Given that approximately 77/1035 (7.5%) of patients present solely with isolated dizziness, [10,15,16,19] we recommended using the FAST4D score with acute onset dizziness/vertigo as a single criterion to suggest stroke, particularly for detecting potentially life-threatening cerebellar infarctions. Given that acute onset dizziness is unmistakably a neurological symptom, labeling it as a stroke merely underscores the urgency of treatment rather than implying misassignment to the specialist department. Hence, it should not overly strain resources.

Building on this, the study by Astasio-Picado et al. (2023) further emphasizes the importance of early symptom recognition and swift action in reducing prehospital delays in stroke care [20]. The FAST4D score has been shown to significantly increase stroke detection rates when utilized by emergency responders. Because of its simplicity, this score could also be effective in raising public awareness of stroke symptoms. A study investigating this potential is already planned.

The median age of patients in this study was 72 ± 15 years, with a nearly balanced distribution between male and female patients (49% to 51%), aligning closely with data reported in other studies on cerebrovascular events [21,22,23]. Analysis of the stroke types within the examined cohort revealed that 93% of the patients experienced cerebral ischemia, and the incidence of intracerebral hemorrhage (ICH) was 7%. This distribution is comparable to that of the general population of stroke patients in Central Europe [21]. As such, the distribution of stroke subtypes within our study population is consistent, suggesting that our observed cohort is representative, thus allowing for meaningful discussion of this study’s primary objectives. Despite original plans for staff training, as outlined in the study protocol, being thwarted by contact restrictions imposed during the COVID-19 pandemic, the rapid implementation of the FAST4D score through the use of pocket cards and occasional consultations in the emergency room was both unexpected and encouraging. Alongside the increased rate of stroke detections, this efficient deployment of the FAST4D score indirectly underscores its feasibility.

This study has several limitations worth noting. Firstly, unlike the new 4D items, the FAST items were not documented individually as positive findings. Consequently, it was not feasible to precisely characterize the positive FAST items in terms of stroke incidence. However, we believe that this limitation did not significantly impact the primary findings of this study. Secondly, this research was conducted exclusively within two stroke units. Despite this constraint, this study included a sufficient number of patients to demonstrate the superior performance of the FAST4D score in this proof-of-concept investigation, particularly when compared with studies that led to the implementation of the BE-FAST mnemonic. Because of the methodology employed, the determination of specificity and positive predictive value was not feasible.

## 5. Conclusions

In summary, supplementing the established FAST score with four additional “D” items—diplopic images, deficit in the field of vision, dizziness/vertigo, and dysmetria/ataxia—significantly enhances stroke detection by approximately 15 percentage points, elevating it from 78% to 93% with the implementation of the new FAST4D score. This improvement also correlates with a notable 65% reduction in false negative diagnoses compared with the conventional FAST score. Consequently, the adoption of the FAST4D score holds promise for enhancing the timely identification and allocation of stroke patients in emergency care settings. Nonetheless, this heightened sensitivity comes at the cost of reduced specificity, leading to over-triage. Nevertheless, this trade-off is deemed necessary to prioritize time-critical interventions for stroke patients, while the associated risks for false positive non-stroke cases are comparatively minimal and generally demanded in the interests of stroke patients.

## Figures and Tables

**Figure 1 jcm-13-05033-f001:**
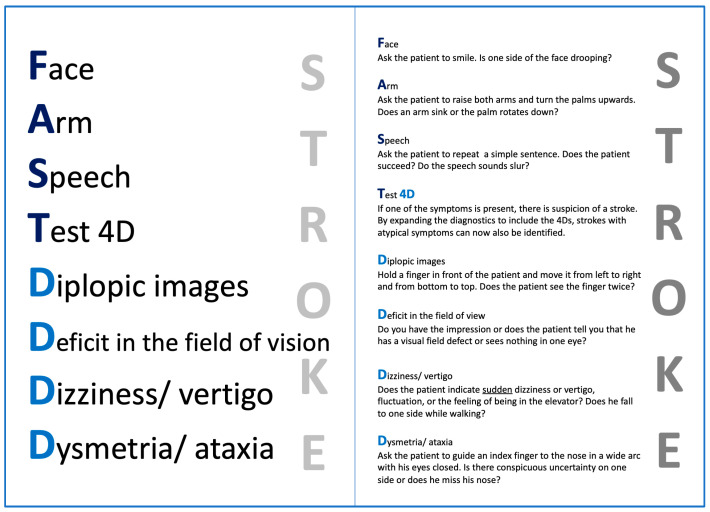
FAST4D pocket card (front and back) for remembering the mnemonic FAST4D, which was handed out to the emergency medical service.

**Table 1 jcm-13-05033-t001:** The four-field panel of the novel FAST4D score compared with the diagnosis of a stroke.

		FAST4D-Positive		
		No	Yes	Total
Stroke	no	0	434	434
	yes	69	966	1035
	total	69	1400	1469

**Table 2 jcm-13-05033-t002:** The four-field panel of the traditional FAST score compared with the diagnosis of a stroke.

		FAST-Positive		
		No	Yes	Total
Stroke	no	224	210	434
	yes	228	807	1035
	total	452	1017	1469

**Table 3 jcm-13-05033-t003:** The frequency and distribution of positive 4D items in patients with a stroke.

		Positive Items					
		Dysmetria/Ataxia	Dizziness/Vertigo	Deficit in the Field of View	Double Vision	Combination of 4D Items	Total
Stroke	no	19	223	3	6	41	292
	yes	68	127	50	11	93	350
	total	87 *	350 *	53 *	17 *	134 *	642

* Significant difference between both groups, McNemar, *p* < 0.05.

## Data Availability

The data analyzed during the current study are available from the corresponding author upon reasonable request.

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
