# Peer review of "FAST4D—A New Score to Reduce Missed Strokes in Emergency Medical Service: A Prospective, Multicentric Observational Proof-of-Concept Trial"

_jcm, 2024, doi:10.3390/jcm13175033_

Round 1

Reviewer 1 Report

Comments and Suggestions for Authors

The manuscript entitled “FAST4D - A New Score to Reduce Missed Strokes in Emergency Medical Service: A prospective, multicentric observational proof-of-concept trial” is well planned and well written, finds FAST4D score increase in stroke detection by 15% and reduces false negatives by 65% by incorporating additional 4 symptoms. This manuscript makes a substantial contribution to the field, offering valuable insights and advancements that enhance our understanding and approach to stroke detection.

Please address below comments

1.      How the 4D can be applied to a patient, who is having any movement disorder like Parkinsons disease?

2.      The study was conducted only in stroke units, which limits the generalizability of the findings.

3.      Please clarify: FAST-4D is how efficient in identifying posterior circulation strokes.

Author Response

Comment 1: How the 4D can be applied to a patient, who is having any movement disorder like Parkinsons disease?

Response 1: Application of FAST4D in Patients with Parkinson's Disease or other Movement Disorders: The fundamental principle for all items in FAST4D is that they must pertain to an acutely onset symptom. Therefore, chronic symptoms of a movement disorder should not influence the score. Admittedly, we do not have information on pre-existing movement disorders within this cohort, and thus cannot make a definitive statement based on our data. However, from the authors' perspective, it is not disadvantageous for the patient to be initially suspected of a stroke when experiencing a worsening of a pre-existing movement disorder. If it is not a stroke, the admitting neurologist should be able to recognize this, and the risk of the emergency services overlooking a stroke as a concurrent condition is minimized.

Comment 2: The study was conducted only in stroke units, which limits the generalizability of the findings.

Response 2: Study Limitations Due to Study Setting: Both stroke units are located in general hospitals that also have internal medicine and surgical departments. This setup allowed us to identify patients who initially tested negative with FAST4D, were subsequently referred to internal medicine or general neurology, and were eventually diagnosed with a stroke (false negatives). This detail is not explicitly stated in our text, so we have added the following sentence to the Methods section: "Patient recruitment primarily took place at the general hospitals in Wetzlar and Friedberg, both of which have a department of neurology, including a certified stroke unit, and a department of general medicine. The stroke units accommodate a comparable number of stroke patients for treatment."

Comment 3: Please clarify: FAST-4D is how efficient in identifying posterior circulation strokes.

Response 3: Sensitivity of FAST4D in Posterior Circulation Strokes: The sensitivity of FAST4D in detecting posterior circulation strokes is 95% (χ² = 0.14; p = 0.71), compared to 55.7% (χ² = 34.2; p < 0.001) for the FAST score. These data are provided in the Supplementary Material for your reference.

Thank you again for your insightful feedback, which has significantly contributed to the improvement of our manuscript

Reviewer 2 Report

Comments and Suggestions for Authors

What type of personnel does EMS consist of? Nurse/Paramedic/Doctor?

What are the key stroke symptoms?

What about other diagnostic tests? MEND, CPSS, LAMS?

Aims very well written.

Is this study unicentric or multicentric?

How far is the hospital where thrombectomies are performed? (How many kilometers?)

Does EMS have a special paper for FAST4D?

The inclusion/exclusion criteria should be explained more clearly

Line 149 statistical, not statisival

I would suggest that the abbreviations be placed next to each other, not below each other

Statistics need to be improved

To improve the manuscript I suggest adding the following articles:

a)        https://doi.org/10.3390/jpm14060596

b)        https://www.ncbi.nlm.nih.gov/pmc/articles/PMC9662151/

c)        https://doi.org/10.3390/medicina58111541

d)        https://doi.org/10.3390/jpm13101519

Comments on the Quality of English Language

Minor English editing

Author Response

Comment 1: What type of personnel does EMS consist of? Nurse/Paramedic/Doctor?

Response 1: The EMS personnel involved in this study primarily consist of paramedics, who undergo a three-year intensive training in emergency medical services. In this EMS region, the management of stroke patients typically does not involve an emergency physician. This system led to faster prehospital care for patients, resulting in quicker access to CT diagnostics and recanalization therapy. In cases of a life-threatening emergency, paramedics can request the additional support of an emergency physician.

Comment 2: What are the key stroke symptoms?

Response 2: The primary symptoms of a stroke, in addition to those covered by the FAST protocol (hemiparesis, speech or language disturbances, and facial paresis), include unilateral sensory disturbances, visual field defects (often resulting from infarcts in the visual cortex, which patients may not notice but can be reliably detected by EMS through simple tests), dizziness (common in cerebellar and brainstem infarcts), ataxia (indicative of cerebellar infarcts), and diplopia (associated with brainstem infarcts). While dysphagia is also a symptom, it is challenging to diagnose accurately even in a clinical setting without speech therapy expertise, making it unsuitable for prehospital screening. Furthermore, there are various brainstem stroke syndromes that typically manifest with one or more of the symptoms mentioned above. FAST4D is considered positive if one or more of these symptoms have occurred acutely.

Comment 3: What about other diagnostic tests? MEND, CPSS, LAMS?

Response 3: The other stroke assessment tools, such as MEND, CPSS, and LAMS, were not compared with FAST4D in this study. We used the FAST score as a benchmark because it has been well established in this EMS region for several years. However, comparing FAST4D with other stroke scales is indeed a valuable direction for future research. It is worth noting that the CPSS is quite similar to the FAST score.

Comment 4: Aims very well written.

Response 4: Thank you for your positive feedback regarding the aims of our manuscript.

Comment 5: Is this study unicentric or multicentric?

Response 5: This was a single-center study involving two Stroke Units and two EMS regions.

Comment 6: How far is the hospital where thrombectomies are performed? (How many kilometers?)

Response 6: The nearest hospital with thrombectomy capabilities is approximately 15 kilometers away from Wetzlar and about 30 km from Friedberg and can be accessed via a highway by ambulance.

Comment 7: Does EMS have a special paper for FAST4D?

Response 7: The EMS initially had a dedicated form listing the FAST4D items. However, due to the COVID-19 pandemic, documentation shifted to a digital format, where the FAST4D items were recorded electronically. Additionally, pocket cards were distributed to all emergency service areas. These cards included a concise version of the FAST4D items, as well as an explanatory text.

Comment 8: The inclusion/exclusion criteria should be explained more clearly

Response 8: Thank you for your feedback. To clarify the inclusion criteria, we have provided both a detailed explanation and a concise list for easier visualization:

“All patients who were either brought to the emergency department (ED) by emergency medical services (EMS) or who presented themselves at the ED with a suspected stroke were included in the data collection. A patient was defined as having a suspected stroke if any of the FAST4D items were positive. This criterion was explicitly stated in the standard operating procedures (SOPs) of both the EMS and the ED. The FAST4D score was documented either by paramedics at the emergency scene as part of the EMS protocol or by the nursing staff in the ED upon admission during the triage process.”

Inclusion Criteria (Concise List):

  1. Patient Presentation
    • Patients brough to the emergency department (ED) by emergency medical services (EMS).
    • Patients who presented themselves at the ED with suspected stroke.
    • Patients who were either brought to the ED by EMS or self-admitted without a suspected stroke, but were later diagnosed with a stroke during clinical evaluation.

  1. Suspected Stroke Definition
    • A patient was considered a suspected stroke if any of the FAST4D items were positive and the symptoms had occurred acutely.
  2. Documentation
    • FAST4D score was documented by paramedics at the emergency scene (as part of the EMS protocol).
    • Alternatively, the FAST4D score was documented by nursing staff in the ED during admission triage.”

Comment 9: Line 149 statistical, not statisival

Comment 10: I would suggest that the abbreviations be placed next to each other, not below each other

Response 9 + 10: In addition to correcting the typographical error in line 149, we have also updated the list of abbreviations according to your suggestions.

Comment 11: Statistics need to be improved

Response 11: The authors are uncertain which areas of the statistical analysis should be improved. This study is a descriptive analysis of a prospective observational study; therefore, confirmatory analyses are inherently not feasible, nor were they intended within this proof-of-concept approach, as is also the case with other prehospital tests (e.g., BEFAST). We described the population because both tests were conducted on the same population, eliminating the need for a group comparison, which our statistician recommended as the most rigorous approach during the planning phase. Subsequently, both tests were evaluated using a 2x2 contingency table, and the test parameters sensitivity and specificity were calculated. Positive and negative predictive values could not be determined because not all patients allocated were captured in the hospitals. Consistent with our overarching goal to improve stroke detection by focusing on the posterior circulation, this aspect was separately analyzed in an additional step, and finally, the occurrence of individual 4D items was presented. We are willing to analyze further aspects if the data allow it; please let us know what the reviewer is missing. Thank you.

Comment 12: To improve the manuscript I suggest adding the following articles:

  1. https://doi.org/10.3390/jpm14060596
  2. https://www.ncbi.nlm.nih.gov/pmc/articles/PMC9662151/
  3. https://doi.org/10.3390/medicina58111541
  4. https://doi.org/10.3390/jpm13101519

Response 12: Thank you very much for your literature suggestions. They have been incredibly helpful in enhancing our manuscript. In particular, we have integrated the article by Astasio-Picado into our discussion and analysis. Your recommendations have significantly contributed to the depth and quality of our work:

„Building on this, the study by Astasio-Picado et al. (2023) further emphasizes the importance of early symptom recognition and swift action in reducing prehospital delays in stroke care. The FAST4D score has been shown to significantly increase stroke detection rates when utilized by emergency responders. Due to its simplicity, this score could also be effective in raising public awareness of stroke symptoms. A study investigating this potential is already planned. “

Thank you for bringing the article by Buleu et al. to our attention. It is indeed a very interesting and valuable resource. Unfortunately, due to the nature of our data collection, we were unable to include in-hospital times in our current analysis, which limits our ability to incorporate this article into the present manuscript. However, this article will be extremely useful for our upcoming confirmatory study, which is already planned as a multicenter trial and will include in-hospital treatment times.

Thank you for highlighting the interesting review by Budincevic et al. This work underscores the diversity of the various stroke assessment scores. We have included this review as an additional reference in our manuscript. Thank you for bringing this valuable source to our attention.

“Various scoring systems have been devised to aid in the recognition of key stroke symptoms [5, 6].”

In addition, we were also able to incorporate the case report by Devlin into our manuscript. It provided valuable insights that complemented our discussion. Thank you for suggesting it.

“This deficiency is particularly notable in posterior circulation strokes, where symptoms are often less specific and may not readily align with typical stroke presentations [8-12].”

Thank you once again for your valuable input, which has contributed to the clarity and accuracy of our manuscript.

Reviewer 3 Report

Comments and Suggestions for Authors

The authors developed a new score to reduce missing strokes in emergency scenarios. As the authors mentioned, the feasibility of applying the new score is questionable, and the cost of training paramedics should be considered, so what is the point of performing neurological exams by non-educated personnel when they should just detect the BEFAST symptoms and transfer the patients to advanced care centers with ED and neurologists who can accurately make the correct diagnosis? As a recommendation to boost the value of the data collected, they should ask paramedics about their confidence in performing a brief neurologic exam and conduct a survey comparing this new score to the traditional BEFAST among paramedics.

Author Response

Comment 1: The authors developed a new score to reduce missing strokes in emergency scenarios. As the authors mentioned, the feasibility of applying the new score is questionable, and the cost of training paramedics should be considered, so what is the point of performing neurological exams by non-educated personnel when they should just detect the BEFAST symptoms and transfer the patients to advanced care centers with ED and neurologists who can accurately make the correct diagnosis?

Response 1: In emergency care, it is crucial to have a simple yet clearly defined score so that EMS personnel, who must also recognize and treat numerous other conditions, can apply it reliably. This is an issue with BEFAST, as it includes fewer items related to posterior circulation, and the balance and eye movement disturbances are not clearly defined, making it difficult for EMS personnel to apply them consistently. The evidence supporting BEFAST is also weak and has highlighted deficiencies in this score, leading to the continued risk of missed strokes.

To clarify this, in the authors knowledge there is one meta-analysis on BEFAST (Chen et al., 2022) that provides limited evidence regarding the effectiveness of BEFAST, as it includes only three studies, two of which were not explicitly tested in the prehospital setting:

  • Aroor et al. (2017): A retrospective study involving 736 patients that examined symptoms not detected by FAST based on patient records, without considering the prehospital setting.
  • Pickham et al. (2019): A prospective study with 359 patients that evaluated BEFAST in the prehospital setting, but showed no significant improvement over FAST.
  • El Ammar et al. (2020): A retrospective study involving 1,965 patients that focused on in-hospital strokes.

Comment 2: As a recommendation to boost the value of the data collected, they should ask paramedics about their confidence in performing a brief neurologic exam and conduct a survey comparing this new score to the traditional BEFAST among paramedics.

Response 2: The approach described by the reviewer aligns with what the authors intend with FAST4D—a high sensitivity to ensure no stroke is missed, with proper evaluation conducted in the hospital by a neurologist to avoid delays due to misallocation. This was demonstrated in the present study and will be further evaluated in a multicenter approach. The FAST4D items are all well-defined, as outlined in the pocket card, and are easy to apply. Additionally, after the analysis, we created a YouTube video to provide a comprehensive and explanatory overview of the assessment process for anyone interested.

We appreciate the suggestion to evaluate the training of FAST4D, and this is already underway, as the authors also find this an interesting area of study. However, these results would not be relevant in the current presentation of test validity, where the primary goal was to demonstrate that the test performs as intended. We do not consider a comparison with BEFAST, a test that, in the view of the authors and others (Pickham, 2019), has not sufficiently demonstrated an improvement in stroke detection compared to FAST, to be meaningful. We prefer to focus on comparisons with FAST and are confident that we have developed a superior test with FAST4D. The authors hope that, after these explanations, the reviewer can better understand why BEFAST, despite its widespread use, is not sufficiently validated to serve as a benchmark for testing a new score.

Round 2

Reviewer 2 Report

Comments and Suggestions for Authors

Accept

Reviewer 3 Report

Comments and Suggestions for Authors

The manuscript was revised, but it did not address my concerns.  "The authors developed a new score to reduce missing strokes in emergency scenarios. As the authors mentioned, the feasibility of applying the new score is questionable, and the cost of training paramedics should be considered, so what is the point of performing neurological exams by non-educated personnel when they should just detect the BEFAST symptoms and transfer the patients to advanced care centers with ED and neurologists who can accurately make the correct diagnosis? As a recommendation to boost the value of the data collected, they should ask paramedics about their confidence in performing a brief neurologic exam and conduct a survey comparing this new score to the traditional BEFAST among paramedics."